# Anti-Inflammatory, Antioxidant, and Neuroprotective Effects of Polyphenols—Polyphenols as an Element of Diet Therapy in Depressive Disorders

**DOI:** 10.3390/ijms24032258

**Published:** 2023-01-23

**Authors:** Anna Winiarska-Mieczan, Małgorzata Kwiecień, Karolina Jachimowicz-Rogowska, Janine Donaldson, Ewa Tomaszewska, Ewa Baranowska-Wójcik

**Affiliations:** 1Institute of Animal Nutrition and Bromatology, University of Life Sciences in Lublin, Akademicka St. 13, 20-950 Lublin, Poland; 2School of Physiology, Faculty of Health Sciences, University of the Witwatersrand, 7 York Road, Parktown, Johannesburg 2193, South Africa; 3Department of Animal Physiology, Faculty of Veterinary Medicine, University of Life Sciences in Lublin, Akademicka St. 12, 20-950 Lublin, Poland; 4Department of Biotechnology, Microbiology and Human Nutrition, University of Life Sciences in Lublin, Skromna St. 8, 20-704 Lublin, Poland

**Keywords:** depression, dietetic polyphenols, anti-inflammatory, antioxidant and neuroprotective effects

## Abstract

Depressive disorders can affect up to 350 million people worldwide, and in developed countries, the percentage of patients with depressive disorders may be as high as 10%. During depression, activation of pro-inflammatory pathways, mitochondrial dysfunction, increased markers of oxidative stress, and a reduction in the antioxidant effectiveness of the body are observed. It is estimated that approximately 30% of depressed patients do not respond to traditional pharmacological treatments. However, more and more attention is being paid to the influence of active ingredients in food on the course and risk of neurological disorders, including depression. The possibility of using foods containing polyphenols as an element of diet therapy in depression was analyzed in the review. The possibility of whether the consumption of products such as polyphenols could alleviate the course of depression or prevent the progression of it was also considered. Results from preclinical studies demonstrate the potential of phenolic compounds have the potential to reduce depressive behaviors by regulating factors related to oxidative stress, neuroinflammation, and modulation of the intestinal microbiota.

## 1. Introduction

Depressive disorders can affect up to 350 million people worldwide, and in developed countries, the percentage of patients with depressive disorders may be as high as 10% [1]. The prevalence of depression in people with various forms of eating disorders is much higher than that of the general population, and in the case of patients with anorexia and bulimia, it ranges from 30% to 90% [2]. Research has shown that in patients suffering from depression, changes in eating behavior related to changes in appetite and food preferences, including avoiding eating specific groups of products and dishes, are more frequent [2]. These changes may lead to severe nutrient deficiencies, which, if they persist for a long time, will negatively impact human health, including the proper functioning of the nervous system.

Despite numerous previous studies, the etiology, pathogenesis, diagnosis, and treatment of depressive disorders are still not fully understood. There are many possible causes of depression. These include: (1) genetic and epigenetic factors [3], (2) abnormalities in the functioning of the hypothalamic-pituitary-adrenal axis (HPA) and the extra-hypothalamic system related to the neurotransmitter corticoliberin (or corticotropin-releasing hormone—CRH) [4], (3) chronic oxidative stress [5,6], and (4) excessive secretion of pro-inflammatory cytokines such as IL-1β and IL-6, which stimulate CRH secretion from the medial nucleus of the hypothalamus, activate the HPA axis, promote the secretion of ACTH and glucocorticoids, and inhibit the serotonergic system [7,8]. A correlation was found between the activation of the HPA axis and the gut microbiota, which has a significant impact on the development of depression. It is believed that the gut microbiota can influence the function of the HPA axis through the activity of cytokines, prostaglandins, and bacterial antigens of various species of microorganisms [7]. Other causes of depression include: deficiency of monoamines—serotonin, dopamine, and norepinephrine—in the central nervous system; abnormalities in specific areas of the brain; decreased activity of the GABAergic system; and inappropriate regulation of the glutaminergic system [9], as well as socio-economic, geographic, dietary, and lifestyle factors [10].

During depression, activation of pro-inflammatory pathways, mitochondrial dysfunction, increased markers of oxidative stress, and a reduction in the antioxidant effectiveness of the body are observed (Table 1). Moreover, the use of exogenous antioxidants reverses these unfavorable physiological conditions [5]. Oxidative stress is manifested by increased lipid peroxidation and the production of mitochondrial reactive oxygen species (ROS) [11], which can damage key biomolecules, such as nucleic acids. Low levels of ROS are necessary for the regulation of the appropriate growth and development of nerve cells, as well as for the long-term potentiation process, through glutamate-dependent mechanisms [12]. Increased levels of oxidative DNA damage and abnormalities in the repair of oxidative DNA damage in the nucleus and mitochondria have been observed in patients suffering from depression [13]. Single nucleotide polymorphisms of genes encoding proteins involved in base excision repair (BER), the major pathway for the removal of oxidative DNA damage, may modulate the risk of depression [14]. In addition to oxidative stress, preclinical and clinical studies have confirmed the effect of increased production of reactive forms of nitrogen (RNS), which lead to nitrous stress and changes in the structure of the brain [15].

It is estimated that approximately 30% of depressed patients do not respond to traditional pharmacological treatments [36]. Nevertheless, drug therapy is the primary treatment for depression. Anti-depressants reduce the production of pro-inflammatory cytokines such as IL-1 or TNF-α (tumor necrosis factor alfa) while increasing the concentration of anti-inflammatory factors such as IL-10 [37]. However, more and more attention is being paid to the influence of active ingredients in food on the course and risk of neurological disorders, including depression [38,39,40,41,42]. The field of nutripsychiatry deals with the treatment and diagnosis of depression in connection with nutrition, where the effects of various nutrients and the nutritional quality of food on mental health are investigated. Food is rich in biologically active compounds, many of which have health-promoting effects. Such compounds include polyphenols, which have strong anti-inflammatory, antioxidant, antibacterial, anti-obesogenic, and neuroprotective effects [11,43,44]. Some preclinical studies indicate that phenolic compounds exhibit anti-depressant properties [40,45]. The possibility of using foods containing polyphenols as an element of diet therapy in depression was analyzed in the review. Based on the information available in the global literature covering the last 10 years, the impact of regular consumption of foods containing polyphenols on the antioxidant status and the occurrence of inflammatory reactions in the body was analyzed. The possibility of whether the consumption of products such as polyphenols could alleviate the course of depression or prevent the progression of it was also considered.

## 2. Information Search Strategy in the Available

The analysis of information available in the global scientific literature was carried out in October 2022. The following databases were used: Scopus, PubMed, Web of Science, and Google Scholar. The databases were searched for the separate or common occurrence of the following keywords: “depression”, “depressive disorders”, “brain”, “polyphenols”, “diet”, “inflammation”, “microbiome”, “oxidative stress”, “antioxidants”, “immunomodulation”, and “epigenetics” in Polish and English. Based on the analysis of titles and abstracts, articles unrelated to the substantive criteria were excluded, and the remaining research and review publications were subjected to detailed analysis in order to select the most relevant publications. The bibliography included in all the selected articles was also analyzed. The search was limited to articles published between 2012 and 2022. Finally, a total of 292 publications were analyzed, of which 219 research papers and 73 reviews were used.

## 3. The Pathogenesis of Depressive Disorders

### 3.1. Inflammation in Depressive Disorders

Inflammation is an important factor/mechanism in the development of depression. Anti-depressants are in fact anti-inflammatory in nature. Activation of inflammatory pathways has been observed in depressed people, as evidenced by increased levels of pro-inflammatory cytokines such as interleukin-1β (IL-1β), IL-6, IL-8, and tumor necrosis factor α (TNF-α) [8,46]. Furthermore, polymorphisms in the IL-1β, TNF-α, and C-reactive protein may increase the risk of depression, and single nucleotide polymorphisms (SNP) in the IL-1β, IL-6, and IL-11 genes may be associated with decreased efficacy of anti-depressant treatment [47,48]. In depressed individuals, there is an increased expression of the NLRP3 inflammasome (NOD-, LRR-, and pyrin domain-containing protein 3), accompanied by elevated serum levels of pro-inflammatory cytokines, IL-1β, and IL-18, and programmed cell death—protein-mediated pyroptosis executive Gasdermin D (GSDMD) [49,50]. The NLRP3 inflammasome (a multimeric protein complex) is activated by several stimuli, including ion flux, mitochondrial dysfunction, ROS production, and lysosomal damage [51]. Activation of inflammasomes that activate inflammatory caspase-1 is the major inflammatory pathway in various types of organism dysfunction [51]. Active caspase-1 cleaves the cytokines pro-interleukin-1β (pro-IL-1β) and pro-IL-18 into their mature and biologically active forms [52]. IL-1β induces the expression of genes controlling fever, pain threshold, vasodilation, and hypotension and facilitates the infiltration of immune cells into infected or damaged tissues [53,54]. In turn, IL-18 is essential for the production of interferon-gamma (IFN-γ), a cytokine that plays a key role in inflammation and immune responses [55]. Removal of superoxide-generating oxidase 2 (NOX2) has been shown to reduce NLRP3 expression in a traumatic brain injury model and disrupt the NLRP3-TXNIP interaction in the cerebral cortex of mice after ischemic stroke, suggesting a tissue-specific role for cytosolic ROS in activating the NLRP3 inflammasome [56]. NADPH 4 oxidase (NOX4) has also been shown to regulate carnitine palmitoyltransferase 1A (CPT1A) and to increase fatty acid oxidation, which contributes to the activation of the NLRP3 inflammasome [57]. In turn, mitochondria are involved in activating the inflammasome primarily through the production of ROS [51]. The likely mechanism of action of the NLRP3 inflammasome is the suppression of DNA repair and the induction of apoptosis through the p53 protein, which may indicate a direct link between inflammation and DNA repair and partly explain the decreased efficiency of DNA repair in people with depression [58,59].

Depressive disorders often coexist with other autoimmune diseases that involve chronic inflammation. This is due to the fact that pro-inflammatory cytokines also act within the central nervous system, where, reaching from peripheral tissues, they are also synthesized de novo by nerve cells, and their activity correlates with the severity of depressive symptoms and the polymorphism of genes encoding serotonin transport [60]. Pro-inflammatory cytokines regulate neurogenesis, which is important for synaptic plasticity, by reducing the survival, proliferation, and differentiation of neural precursor cells [61]. Activation of signaling pathways that lead to the activation of pro-inflammatory genes reduces cell regeneration in important migration pathways in the brain [62]. Higher levels of IL-8 have been found in the cerebrospinal fluid of patients with unipolar depression than in healthy patients, indicating the importance of these pro-inflammatory biomarkers in the pathophysiology of depression [63]. Changes in immune system activity in depression, including both activation and suppression at the same time, may suggest an immune imbalance in depression [37].

The kynurenine pathway is probably the link between generalized inflammation in the body and depressive symptoms [64]. In plasma, lower levels of picolinic acid, higher levels of quinolinic acid, and reduced levels of neuroprotective to neurotoxic metabolite ratios were found in depressive patients compared to the healthy controls. In the cerebrospinal fluid, a significantly lower level of picolinic acid was found in depressed patients compared to healthy subjects [64]. This pathway metabolizes tryptophan, an amino acid that is a precursor to neurotransmitters such as serotonin and melatonin. Inflammation causes excessive activation of indoleamine-2,3-dioxygenase (IDO), an enzyme that converts tryptophan to N-formylkynurenine, which is then metabolized to kynurenine [64]. Excessive activation of IDO causes the availability of tryptophan to decrease, which has consequences during the synthesis of neurotransmitters. In addition, kynurenine is then metabolized to 3-hydroxykynurenine, 3-hydroxyanthranilate, or quinolic acid, which are cytotoxic [65]. Moreover, due to these alterations in the metabolic pathways, the reduced amount of tryptophan limits the biosynthesis of serotonin and, consequently, may be the cause of increased susceptibility to depressive mood [66]. It was found that the concentration of tryptophan is negatively correlated with the levels of pro-inflammatory cytokines, positive acute phase proteins (CRP), and neopterin produced by active monocytes [67].

Neurotransmitters regulate the secretion of cytokines through the level of cortisol. For example, acetylcholine, dopamine, and norepinephrine all promote the secretion of the corticotropin-releasing hormone (CRH) in the hypothalamus, while serotonin inhibits the secretion of CRH in the hypothalamus and adrenocorticotropic hormone in the pituitary gland [46]. When cortisol levels are low, the production of pro-inflammatory cytokines is increased, while their production is inhibited by high levels of cortisol [46]. In pathological conditions such as acute or chronic inflammation or tissue damage, the immune system and macrophages are activated to increase the level of pro-inflammatory cytokines. Inflammation markers, including IL-6, TNF-α, TNF-β1, IFN, and CRP have been shown to be constantly increased in depressed patients [68].

Depression is associated with polymorphisms in inflammation-related genes, while several gene variants are involved in both immune activation and depression [48]. Studies conducted in a group of 190 depressed people and 100 healthy people showed that the expression of PON2 and PON3 genes at the protein level was significantly higher in patients with depression, while the mRNA expression of PON1, PON2, and PON3 genes did not differ in patients with depressive disorders compared to the control group. It was also shown that the expression of the MPO gene, both at the mRNA and protein levels, was significantly lower in patients with depressive disorders than in the control group [69].

### 3.2. Microbiota and Systemic Inflammation in Depression

Currently, the importance of the microbiota and its differentiation as a trigger for generalized inflammation is recognized [70,71]. The destabilization of the composition of the intestinal microbiota by antibiotics resulted in a 20–50% increase in the risk of depression [72]. Moreover, alpha diversity was found to be negatively associated with depressive symptoms, while beta diversity showed a significant association with major depressive disorder, psychosis, and schizophrenia [73,74]. Depression has also been linked to insufficient numbers of *Firmicutes* in the intestines [73]. The optimal composition of the intestinal microflora determines the efficient functioning of the immune system. It has been shown that the microbiota of people with depressive disorders is characterized by lower levels of *Faecalibacterium* and *Coprococcus* and higher levels of *Eggerthell* compared to that of healthy people, suggesting that the microbiome of people with depression contains less anti-inflammatory butyrate-producing bacteria and higher numbers of pro-inflammatory-producing bacteria [74]. Other studies have shown that the stool of people with depression contained more *Bilophil* (type *Proteobacteria*) and *Alistipes* (type *Bacteroidetes*), and less *Anaerostipes* and *Dialister* (type *Firmicutes*) compared to that of healthy people [75,76]. It should be noted that *Bilophila* and *Alistipes* are gram-negative bacteria, and thus the lipopolysaccharides that form part of their membranes can stimulate the innate immune system by activating TLR-4 [77]. Activation of TLR-4 induces depression-like behavior in animal models and has been proposed as a key factor in the inflammatory theory of depression [78]. It is also possible that *Alistipes* may affect the availability of tryptophan in the body, which is necessary for the synthesis of the neurotransmitter serotonin, which in turn may upset the balance of the serotonergic system [79].

Due to the impaired functioning of the intestinal barrier, bacteria migrate from the gastrointestinal tract, which activates the cells of the immune system and affects the functioning of the immune, endocrine, and nervous systems [80,81,82]. Certain factors negatively affect the functioning and selectivity of the intestinal barrier, including increased concentrations of IL-1β, IL-6, TNF-α, IFN-γ, and NF-κB, increased production of ROS and nitric oxide (NO), and decreased concentrations of exogenous antioxidants [83]. Gut barrier dysfunction increases the potential influx of antigens, inflammatory cytokines, T-cells, and macrophages into the brain, triggering neuroinflammation through the activation of microglia and astrocytes [84]. In people with depression, the dysfunction of the intestinal barrier is accompanied by inflammation, and the degree of intestinal barrier dysfunction correlates with the severity of their depressive symptoms [84]. Increased concentrations of immunoglobulins IgA and IgM against the lipopolysaccharides of bacteria in the microbiome have been observed in patients with depression [85]. In turn, increased translocation of lipopolysaccharide from Gram-negative enterobacteria activates the inflammatory response system, which leads to an increase in the level of pro-inflammatory cytokines (IL-6, IL-1β, TNF-α) and the mitogen-induced lymphocyte response [86]. Stress negatively affects the formation and diversity of the intestinal microflora [87]. On the other hand, microorganisms belonging to the group of psychobiotics (probiotics and prebiotics that confer mental health benefits through interactions with commensal gut bacteria) produce neurotransmitters, including gamma-aminobutyric acid, serotonin, dopamine, and short-chain fatty acids (SCFA; acetic, propionic, and butyric), which directly affect the nervous system [70,88,89]. SCFAs, i.e., the main metabolites produced in the colon by gut microbiota, are transported via blood vessels to the brain, where they modulate functions of neurons, microglia, and astrocytes and affect the blood-brain barrier (BBB) [80,86]. Elevated levels of toxins and/or microbes may alter the functioning of the BBB, which may lead to neurodegeneration [80]. In this context, the influence of the microbiome-gut-brain axis on the stress response of the HPA axis in alcohol-induced depression is known [90].

### 3.3. Oxidative Stress in Depressive Disorders

At the site of inflammation, as a result of incomplete reduction of oxygen, among others, a superoxide anion is formed, which influences the development of oxidative stress. The superoxide anion is a reactant that produces a very reactive hydroxyl radical [91]. It is also a link between oxidative stress and nitrosation stress because, as a result of the reaction of this radical with NO, superoxynitrite is formed, causing nitration of cell elements [15,92]. The properties of proteins modified in this way are altered, thus making them unable to fulfill their functions appropriately. Consequently, cell signal transduction may be disturbed, leading to cell death. As a result of the increased synthesis of NO by cells of the immune response in the inflammatory focus, not only does the pool of ROS increase but also that of reactive forms of nitrogen (RNS), which is referred to as nitrosative stress [92]. Nitrosation stress is associated with the synthesis of an excessive amount of molecules containing a nitrogen atom as part of their structure, such as nitrogen oxide (NO) and its derivatives: nitroxyl ions (NO=), nitric acid (III) (HNO_2_), nitric acid (V) (HNO_3_), nitric oxygen radicals (•NO), nitric dioxide radicals (•NO_2_), nitrogen dioxide (NO_2_), nitric oxide (III) (N_2_O_3_), and a particularly reactive superoxide anion—peroxynitrite (ONOO−) [93]. The role of neuronal NO synthase (nNOS) in the pathophysiology of depression has been demonstrated, and in the case of antidepressants, it has been shown that they reduce NO levels in the serum of patients with depression [92]. Studies on mice have shown that lowering NO levels in the brain and serum effectively reduces symptoms of depression [94,95].

Oxidative stress is described as a condition in which cellular antioxidant defense is inadequate due to the over-release of ROS [11]. Consequences of oxidative stress include fragmentation of lipids and/or structural changes, protein denaturation, disturbances in DNA replication, deformation of cell organelles, and consequently whole cells [11,96]. ROS-induced oxidative stress leads to inflammation and also triggers NF-κB dependent (nuclear factor kappa-light-chain-enhancer of activated B cells) gene transcription for many pro-inflammatory factors [97]. It is presumed that oxidative stress, along with nitrosative stress, may have an impact on the pathophysiology of depression [15]. The brain is an organ that is particularly exposed to oxidative damage. This is due to the high oxygen consumption, high lipid content, and relatively low content of antioxidant enzymes in the brain [91]. In contrast, increased oxygen consumption results in increased production of ROS and RNS in the brain. Particularly significant changes in the activity of antioxidant enzymes can be observed in the mitochondria of the brain, which are the main source of the superoxide radical anion (O_2_•−) and hydrogen peroxide (H_2_O_2_) [98]. Mitochondria are the main source of endogenous ROS and are therefore particularly vulnerable to disrupting antioxidant and detoxification systems [99]. It is worth noting that in people with depression, there is a higher production of mitochondrial ROS and a decrease in ATP production, as well as a reduced level of coenzyme Q10, which is an essential component of the respiratory chain, and its reduced amount indicates mitochondrial dysfunction [99,100]. Adequate ATP levels are especially important for the brain, which consumes a lot of energy but is unable to store large amounts of it, among other things, because neurons do not store glucose [101]. Inhibition of the activity of complexes I, III, and IV of the respiratory chain, as well as creatine kinase, was observed in the brain cortex and cerebellum of a rat model of depression [102].

The main targets of ROS in the brain are lipids, especially phospholipids, and their excessive peroxidation in the brain is an important event in the pathogenesis of depression [103,104,105]. Moreover, studies in mice with chronic depression have shown that persistent changes in brain peroxidation occur not only during anhedonia but also during recovery [105]. Brain lipids determine the location and function of proteins in the cell membrane and thus regulate synaptic capacity in neurons; they can also act as messengers [106]. The results of preclinical studies suggest a key role for film-forming n-3 polyunsaturated fatty acids, glycerolipids, glycerophospholipids, and sphingolipids in the induction of behaviors related to depression and anxiety [106]. Possible mechanisms may include increased production of pro-inflammatory cytokines, which can activate the HPA axis, and alteration of membrane fluidity, which affects membrane enzymes, ion channels, receptor activity, and neurotransmitter binding [106]. It has been suggested that an increased ratio of omega-6 to omega-3 fatty acids in cell membranes is involved in the pathogenesis of depression [107]. People with depressive disorders have an increased amount of malondialdehyde (MDA), which is a marker of oxidative damage to fatty acids [108]. In the study by Stefanescu and Ciobic [109], the level of MDA was higher in patients with subsequent depressive episodes than in patients who developed the disease for the first time. Another marker of oxidative damage to fatty acids such as arachidonic acid is 8-iso-prostaglandin F2, an increased amount of which has been detected during the course of depression [110].

Reduced amounts of non-enzymatic antioxidants such as vitamins A, C, and E, albumin, coenzyme Q10, zinc, and glutathione have been detected in the blood, plasma, and brain of people with depression [5], which proves a reduction in the effectiveness of antioxidant defense in these people. Moreover, in people with depression, there are disturbances in activity and a decrease in the expression of enzymatic antioxidants, such as superoxide dismutase (SOD), catalase (CAT), and enzymes related to glutathione metabolism: peroxidase, reductase, and S-transferase [24,111]. Some polymorphs of manganese superoxide dismutase (MnSOD, SOD2) are more common in people with depression than in healthy people, which may lead to reduced MnSOD uptake by mitochondria and cause instability of this enzyme mRNA [112,113]. The genetic polymorphism of MnSOD (rs4880) has also been shown to have no effect on 6-month antidepressant treatment and inflammatory biomarkers in depressed patients, suggesting that MnSOD is not the major genetic determinant of the antidepressant response [113]. Studies have shown the presence of more 8-oxoguanine (8-oxoG), a marker of oxidative DNA damage, in depressed patients compared to healthy subjects [114]. Moreover, it has been proven that DNA damage present in depression may be caused by disturbances in DNA repair, which are slower in the cells of depressed patients [114].

## 4. Anti-Inflammatory, Antioxidant, and Neuroprotective Effects of Polyphenols

Polyphenols (flavonoids and non-flavonoids, such as resveratrol, curcumin, coumarin, and phenolic acids) exhibit anti-inflammatory, antioxidant, and neuroprotective properties (Figure 1) and are thus considered complementary medicinal compounds in the treatment of mental disorders. The anti-inflammatory properties of polyphenols result from their: (1) effects on immune cells; (2) inhibition of the secretion of pro-inflammatory cytokines; (3) influence on the expression of genes responsible for the synthesis of pro-inflammatory cytokines; and (4) induction of apoptosis, which reduces DNA damage [115,116,117]. Phenolic compounds exhibit antioxidant properties due to their ability to: (1) scavenge ROS; (2) reduce the production of ROS by inhibiting the activity of oxidative enzymes and chelating trace elements; (3) increase the activity of endogenous antioxidants; and (4) donate an electron or hydrogen atom, which facilitates the neutralization of singlet oxygen [11,91,118,119]. The neuroprotective effect of polyphenols is due to their antioxidant and anti-inflammatory properties, which include (1) the modulation of multiple neurotransmitter systems, (2) modulation of the functions of the HPA axis and intracellular signaling pathways involved in neurogenesis, neuroplasticity and cell survival, (3) the prevention of demyelination and neurodegeneration, (4) the ability to increase cerebral blood flow by stimulating NO production in the endothelium, (5) the induction of sirtulin-1/AMP-activated protein kinase (SIRT1/AMPK), which reduces microglia activation, and (6) the inhibition of key signaling pathways in activated microglia such as NF-κB, mitogen-activated protein kinases (MAPK) and Janus kinase/signal transducers and activators of transcription (JAK-STAT) [120,121,122,123,124].

## 5. Foods Containing Polyphenols as Part of Diet Therapy in Depression—A Research Review

Polyphenols are naturally occurring compounds; they are secondary metabolites of plants. Fruits, vegetables, cereals, and such beverages as tea represent the main sources of polyphenols (Figure 2). A relationship has been shown between the way people eat and the risk of depression, anxiety, and stress disorders [125,126,127]. The total antioxidant capacity of the diet consumed is inversely correlated with depression and some biomarkers of oxidative stress in menopausal, postmenopausal, and diabetic women [128,129,130], as well as in young women and girls [131,132,133]. The study by de Oliveira et al. [130] performed in Brazil showed that depressed women consumed fewer polyphenols than healthy women, which may indicate the importance of this group of food ingredients in the prevention of depressive disorders. A study in a group of 50 people found that an 8-week high-antioxidant diet containing blueberries and dark chocolate reduced symptoms of depression [134]. Similar results were obtained by Huang et al. [135]. Traditional Japanese and Norwegian diets rich in fruit and vegetables have also been linked to a reduced risk of depression [136]. Currently, the diet with the most evidence that it offers protection against the risk of depression is the polyphenol-rich Mediterranean diet, which has been recognized as a promising treatment strategy for depression [137,138]. Polyphenols may inhibit the expression and function of pro-inflammatory cytokines, transcription factors, and protein complexes that trigger neuroinflammatory responses and thus may diminish depressive behavioral symptoms [124]. Changes in the microbiome have been observed in depressed people who consume flavonoid-rich orange juice; these changes have been linked to improved health in these people [139]. Another study found quercetin to ameliorate lipopolysaccharide-induced depression in rats [140]. On the other hand, the relationship between the content of non-enzymatic antioxidants in the diet and symptoms of depression in Japanese workers was investigated, but no significant correlations were found [141], although the healthy Japanese lifestyle—characterized by the consumption of large amounts of vegetables, fruits, mushrooms, and soy products—is inversely associated with symptoms of depression [142]. In other studies, polyphenol supplementation was found to be effective in improving health in depression, but phenolic compounds do not work equally; therefore, it was suggested that in diet therapy, the type of polyphenols should be selected individually for the patient [143]. It has also been confirmed that Western eating styles (low in fruits and vegetables, high in fat and saturated fatty acids, sugar, sodium, and processed food) can increase the risk and severity of depression in adolescents [144].

### 5.1. Influence of Polyphenols on Inflammation in Depression

Persistent inflammation disrupts several molecular and cellular pathways in the central nervous system, associated with the pathophysiology of depression [124]. It has been shown that the remission of depressive states is associated with the normalization of the levels of inflammatory markers in the body [145]. Numerous preclinical, clinical, and laboratory animal studies (Table 2 and Table 3) have shown the anti-inflammatory effect of polyphenols in depressive disorders. Polyphenols inhibit the MAPK signaling pathway, mediating oxidative stress and inflammation in depression, and they also regulate NF-κB activation [146].

#### 5.1.1. Quercetin

Quercetin, a flavonol present in fruits, vegetables, and some herbs, has been proven to have anti-depressant, anti-cancer, anti-bacterial, anti-oxidant, anti-inflammatory, and neuroprotective effects [182]. Quercetin can regulate the level of neurotransmitters in the body, promote the regeneration of hippocampal neurons, and improve HPA axis dysfunction [182]. In addition, quercetin crosses the blood-brain barrier and is present in the brain just a few hours after administration, thanks to which it has an effective neuroprotective effect [183]. Studies in mice have shown that quercetin protects against stress-induced anxiety and depressive behaviors and improves memory by regulating serotonergic and cholinergic neurotransmission [184]. Quercetin may ameliorate lipopolysaccharide-induced depression-like behavior and impairment of learning and memory in rats, which may be due to the regulation of an imbalance in copin 6 expression and the triggering of receptors expressed on myeloid cells (TREM1/2) in the hippocampus and prefrontal cortex [185]. A study conducted on rats showed that the combination of quercetin supplementation and physical training exerts a strong anti-cancer and antidepressant effect by suppressing inflammation and regulating the brain-derived neurotrophic factor (BDNF) axis—tyrosine kinase β receptor (TrKβ)—β-catenin in the prefrontal cortex [186]. It has also been shown that the synergistic effects of quercetin and curcumin are particularly effective as antidepressants. In rats exposed to carrageenan, the simultaneous administration of quercetin and curcumin was found to modulate the expression of heme oxygenase-1 mRNA and TNF-α, confirming the anti-inflammatory effect of these compounds, as well as the reduction in oxidative stress [187]. A 0.8% quercetin supplementation reduced levels of interferon γ, IL-1α, and IL-4 in male C57Bl/6j mice [188]. In turn, the administration of 10 mg/kg of quercetin to obese Zucker rats reduced TNF-α production in adipose tissue [189].

#### 5.1.2. Curcumin

Curcumin, a naturally occurring biologically active compound derived from Curcuma longa, exhibits a wide range of pharmacological properties and has been recognized as a potent antidepressant with multiple mechanisms including monoaminergic imbalances (related to serotonin, dopamine, norepinephrine, and glutamate), effects on neurotransmitters, neuroprogression, the HPA axis, dysregulation of inflammatory and immune pathways, oxidative and nitrosative stress, and mitochondrial disorders [190]. Curcumin reduces chronic mild stress induced depression symptoms and memory deficits by modulating oxidative stress and inhibiting acetylcholinesterase activity [191]. Curcumin has also been shown to reduce the expression of inflammatory cytokines TNF and IL-1, adhesion molecules such as ICAM-1 (intercellular adhesion molecule-1) and VCAM-1 (vascular cell adhesion molecule-1), and inflammatory mediators such as prostaglandins and leukotrienes [192]. Curcumin also inhibits the synthesis of enzymes involved in inflammation, such as COX (cyclooxygenase), LOX (lipoxygenase), MAPK (mitogen-activated protein kinase), and IKK (kappa kinase inhibitor), lowers the levels of NF-κB and STAT3 (signal transducer and activator of transcription 3), reduces the expression of TLR-2 (toll-like receptor-2) and TRL-4, and increases the synthesis of PPARγ (peroxisome proliferator-activated gamma receptor) [193,194,195]. It has also been shown that curcumin is able to enhance the anti-inflammatory and phagocytic effects in microglia cells, showing a direct regulatory effect on Aβ42 peptide phagocytosis as well as weakening the inflammatory image on PGE2-stimulated N9 cells [196]. Studies in Wistar Kyoto rats provided further evidence of the antidepressant effects of curcumin, possibly by increasing neurotrophic activity in the hippocampus [197]. Studies have shown that curcumin is safe, well-tolerated, and effective in depressed patients [198,199].

#### 5.1.3. Tannic Acid

Tannic acid shows strong anti-inflammatory [200] and antioxidant properties [77,96,201,202], as shown in studies carried out on laboratory animals. Tannic acid can regulate many signaling pathways, including those related to TGF-β (transforming growth factor β), EGFR (estimated GFR), and bFGF (basic fibroblast growth factor) [203]. The administration of tannic acid significantly inhibited the levels of TNF-α, IL-1β, ET-1, and NF-κB in rats exposed to doxorubicin [204]. Tannic acid has been shown to be a nonselective monoamine oxidase inhibitor and therefore increases the levels of monoaminergic neurotransmitters in the brain [205]. Studies in mice have shown a positive effect of tannic acid on lipopolysaccharide-induced depressive and inflammatory lesions [206]. On the other hand, in stressed dogs, a decrease in the level of pro-inflammatory interleukins was found after the use of tannic acid [207].

#### 5.1.4. Chrysin

Chrysin (5,7-dihydroxy flavone), a flavonoid isolated from plants such as Passiflora coerulea, Passiflora incarnata, and Matricaria chamomilla [208], has very strong anti-inflammatory and antioxidant properties [209]. Chrysin has been shown to exert neuro-pharmacological effects by activating neurotransmitter systems (GABAergic, serotonergic, dopaminergic, and noradrenergic), eurotrophic factors (e.g., brain-derived neurotrophic factor and nerve growth factor), and some signaling pathways. Chrysin also alters serotonin levels and modifies the expression of its receptors, including 5-HT1A and 5-HT2A in the raphe nucleus and hippocampus [209,210]. It also regulates the levels of dopamine and norepinephrine in the central nervous system [208]. Chrysin acts in brain structures involved in the pathophysiology of anxiety and depressive disorders, such as the hippocampus, prefrontal cortex, raphe nucleus, and striatum [208]. Both clinical and laboratory animal studies and isolated cell cultures have shown a reduction in inflammatory markers (TNF-α, IL-1β, IL-6, NF-κB, IKK-β) following the use of chrysin [159,208,211,212,213,214].

#### 5.1.5. Peoniflorin

Peoniflorin, a polyphenolic compound found in Radix Paeoniae Alba (Paeonia lactiflora), used in Chinese medicine, has antidepressant activity, although the potential therapeutic mechanism has not been thoroughly investigated [215]. Peoniflorin has been shown to reduce depressive symptoms in rats and correct an abnormal metabolic profile [215]. This study demonstrated that the metabolites critical for peoniflorin function are citric acid, thiamine monophosphate, gluconolactone, 5-hydroxyindole acetic acid, and stachyose, targeting SLC6A4, TNF, IL6, and SLC6A3. However, a particularly important metabolic pathway is the citrate cycle. Paeoniflorin alleviates spatial learning impairment in mice subjected to chronic unpredictable mild stress, which causes depression symptoms [216]. In turn, the elimination of changes in the hippocampus and the improvement of neuroplasticity in the CA1 region proved the neuroprotective effect of peoniflorin [216]. Peoniflorin lowers TLR4, NF-κB and NLRP3 levels, inhibits TLR4/NF-κB/NLRP3 signaling, and reduces proinflammatory cytokines and microglia activation in the hippocampus of lipopolysaccharide-induced mice [217]. By inactivating microglia, it also increases the secretion of neuronal fibroblast growth factor 2 (FGF-2), an anti-inflammatory factor involved in the regulation of the proliferation, differentiation, and apoptosis of neurons in the brain [217].

#### 5.1.6. Tea Polyphenols

Regular tea consumption has been found to have an antidepressant effect, which is related to its phenolic compound content [218]. In Korean studies, people who consume more than three cups of green tea a week have a 21% lower incidence of depression [219], and increasing tea consumption by three cups a day was associated with a 37% reduction in the risk of depressive disorders [220]. Green tea polyphenols increase the Nrf2 (nuclear factor erythroid-2-related factor 2) signaling pathway and suppress oxidative stress and inflammatory markers [221]. With increased HPA axis activity in depression, the hippocampus becomes sensitive to a constant, elevated level of corticoliberin, the corticotropin-releasing hormone (CRH), which in turn causes the body to release more cortisol in an attempt to regulate CRH production [218]. In turn, the increased level of cortisol causes greater stress and intensifies inflammation (secretion of increased amounts of NF-κB, TNF-α, IL-1, IL-2, IL-6, and monocyte chemoattractant protein-1 MCP-1), which leads to the release of more cortisol and, eventually, causes neuropathology. Over time, chronically elevated levels of stress hormones worsen important neural networks in the brain, including the monoaminergic and limbic systems, and reduce hippocampal volume and neuroplasticity [218]. Additionally, inflammation coincides with a significant decrease in serotonin levels, and both serotonin synthesis and structure depend on tryptophan as a precursor [218]. The immune system tries to stop bacterial growth by regulating the metabolic pathway of tryptophan to kynurenine through the enzyme indoleamine 2,3-dioxygenase. Inflammatory cytokines induce indoleamine activity, causing a significant decrease in tryptophan levels and a corresponding decrease in serotonin synthesis, and low levels of serotonin resulting from inflammation exacerbate the symptoms of depression. Moreover, inflammatory cytokines can elevate cortisol levels and activate the HPA axis, causing chronic stress [218]. The major ERK/CREB/BDNF signaling pathway associated with depression is stimulated by tea polyphenols, primarily theaflavin, theanine, epigallocatechin gallate (EGCG), and combinations of catechins and their metabolites. Theaflavin and EGCG are potent anti-inflammatory agents that act by downregulating signaling NF-κB [218]. Wang et al. [222] suggested that theanine may participate in the relationship between inflammatory cytokines and the HPA axis. Catechins are active ingredients in green tea. The antidepressant effect of EGCG was observed in a rat model of chronic unpredictable mild stress-induced depression, where EGCG inhibited neuroinflammation (reduced levels of IL-6 and NO) in the hippocampus [223]. Intraperitoneal administration of EGCG (25 mg/kg) for 14 days prior to a single prolonged stress was able to reduce IL-1β and TNF-α levels in the hippocampi of mice, which was associated with improved cognitive abilities and object recognition memory during behavioral tests [224]. Gallocatechin (GCG) and EGCG were found to inhibit lipopolysaccharide-induced p65 phosphorylation and showed a similar ability to regulate NF-κB activation in vitro [225]. In mice with lipopolysaccharide neuritis, theaflavin black tea suppressed inflammatory cytokine production and significantly reduced depression-like behavior [226]. The results of the cited study indicate that the anti-inflammatory activity of theaflavin on microglia is stronger than that of common polyphenols but comparable to EGCG. In vivo studies, on the other hand, revealed that theaflavin-3,3′-digalate (TFDG) significantly inhibited lipopolysaccharide-induced expression of inflammatory biomarkers such as TNF-α, IL-1β, and IL-6 [227,228]. Ethanolic oolong tea extract showed comparable anti-inflammatory activity to EGCG by reducing several inflammatory responses in a lipopolysaccharide-induced mouse macrophage cell line [229]. The anti-inflammatory effects of tea and tea polyphenols have also been demonstrated in other diseases [230,231,232].

### 5.2. Influence of Polyphenols on the Microbiome

The studies emphasize the impact of polyphenols on intestinal health, resulting from their ability to (1) modulate intestinal barrier function, (2) innate and adaptive immune responses, (3) signaling pathways, and (4) modify the composition of the intestinal microbiota [233,234]. Since a large proportion of polyphenols (90–95%) remain unabsorbed in the gastrointestinal tract, they can accumulate in the large intestine, where most of them are extensively metabolized by the intestinal microbiota [233]. Polyphenols can modulate the mucus barrier, nutrient uptake, and viscoelastic microenvironment of the gut bacteria, as demonstrated by ellagic acid and resveratrol studies in goblet cells in the colon mucosa in a rat model of Crohn’s disease [235,236]. Phenolic compounds also affect the intestinal microbiome by inhibiting the development of harmful bacterial species or stimulating the growth of certain species [71,237]. The microbiome can also metabolize food compounds and produce bioactive molecules [237,238]. Phenolic compounds show selective bactericidal activity by damaging bacterial cell membranes. Their inhibitory effect on the development of *Bacillus cereus, Campylobacter jejuni, Clostridium perfringens, Escherichia coli, Helicobacter pylori, Legionella pneumophila,* and *Mycobacterium* spp. has been proven [239]. An in vitro study using human fecal microbiota showed that different polyphenols (in this case EGCG and TFDG) undergo different microbial transformations, but their modulating effect on the intestinal microbiota is similar and mostly related to the growth-promoting activity of *Bacteroides, Faecalibacterium, Parabacteroides,* and *Bifidobacterium* and inhibitory *Prevotella* and *Fusobacterium* [239]. In this regard, research suggests that polyphenols are candidates for prebiotic-like compounds. Prebiotics are non-living gut microbial stimulants [71].

Interactions between polyphenols and the gut microbiota lead to changes in the composition of the microbiota and the production of metabolites, including SCFA, which exert biological effects both locally and systemically. As a result of the cleavage of glycosidic bonds in polyphenols, glycans are formed, which are an important nutrient for the intestinal microbiota, especially for *Bacteroidetes* [240]. In vitro studies have shown that tea polyphenols promote the growth of *Bacteroides, Faecalibacterium, Parabacteroides,* and *Bifidobacterium* and inhibit *Prevotella* and *Fusobacterium*; therefore, they are considered probiotic substances [237,241,242,243]. Probiotics are lactic acid bacteria. The main goal of supplementing the diet with probiotics is to increase the population of beneficial bacteria and eliminate pathogens [71]. Studies carried out on C57BL/6J mice showed a reduction in the number of Firmicutes in the caecum and an increase in the number *of Bacteroidete* due to the production of polyphenols extracted from green and black tea, as well as an increase in SCFA synthesis, which was correlated with an increase in the number of *Pseudobutyrivibrio* [244]. Similarly, in studies by Sun et al. [245] and Guo et al. [243], tea polyphenols inhibited the growth of *Bacteroides-Prevotella* and *Clostridium histolyticum* and stimulated the growth of *Bifidobacterium, Lactobacillus,* and *Enterococcus* bacteria and stimulated SCFA synthesis. Oolong tea polyphenols used in obese mice after 4 weeks caused an increase in the biodiversity of bacteria and the number of bacteria types producing butyrate and acetate, primarily *Bacteroidetes*, with a simultaneous favorable decrease in the proportion between *Firmicutes* and *Bacteroidetes* [242]. The oolong tea polyphenols showed the ability to regulate circadian rhythms by enhancing beneficial intestinal microbiota and regulating gene expression, which influences metabolic pathways [243]. Tea polyphenols may have an effect on the microbiome by maintaining optimal redox status [246]. In this study, *Lachnospiraceae, Bacteroides, Alistipes,* and *Faecalibaculum* were identified as biomarkers of gut redox status. Flavonols belonging to phenolic compounds regulate the adhesion of bacteria to the intestinal walls, this mainly applies to *Lactobacillus acidophilus* LA-5 and *Lactobacillus plantarum* IFPL379 [247]. Kaempferol present in tea leaves improves the integrity of the intestinal barrier and inhibits inflammation in the intestines by reducing the activation of the TLR4/NF-κB pathway [248]. Studies on laboratory animals and in humans have shown that catechins inhibit the growth of pathogenic bacteria, *Clostridium difficile* and *Staphylococcus* spp., and stimulate the growth of beneficial *Bifidobacterium* bacteria [240,249,250,251]. They also improve the integrity of the intestinal barrier and reduce pro-inflammatory reactions, which have been demonstrated in studies on laboratory animals and in humans [240,249,250,251]. Studies on rats with colitis showed that the EGCG present in green tea stimulates an increase in *Akkermansia* abundance and the production of SCFA [252].

The participation of microbiota in the absorption of polyphenols is of great importance for human health because, due to their complex structure and high molecular weight, they are characterized by low bioavailability in the small intestine [241]. In the large intestine, they are converted into bioactive, small-molecule phenolic metabolites by the intestinal microbiota [43]. Although theaflavin and its galloyl derivatives, as well as teasinensin A, are more resistant to degradation by intestinal bacteria [241,253], in vitro studies have shown that the microbiota play an important role in the metabolism of theaflavins in both mice and humans. Lactobacillus plantarum 299v and Bacillus subtilis are of particular importance [254].

It is also recommended that products containing polyphenols be included in the diet under certain conditions. A good example is yoghurt. It is enriched with plant extracts, it makes a very good source of polyphenols and other antioxidants, and may also contain probiotic bacteria that benefit the intestinal microbiome [255].

### 5.3. Influence of Polyphenols on Antioxidant Status in Depression

Polyphenols can interact with ROS and prevent oxidative damage to tissues [146]. ROS causes oxidative stress, lipid peroxidation, protein oxidation, and DNA damage in neuronal tissues. Phenolic compounds may play an important role in the treatment of mental illness due to their ability to protect neurons from oxidative stress, ameliorate ischemic damage by inhibiting lipid peroxidation, and their ability to interact with NO from the vascular endothelium and reduce inflammation [256]. The oxidative stress-induced pathophysiological processes in nervous tissues are recognized as one of the leading mechanisms in the induction of depression.

Numerous studies have shown that a diet rich in phenolic compounds lowers oxidative stress levels. Polyphenols are an important part of the diet as they are present in many natural foods, herbs, and spices. Increased consumption of fruit and vegetables has been shown to reduce the risk of depression in various groups of people [257,258]. A 100-g increase in fruit and/or vegetable consumption was associated with a 3% reduction in depression risk [259]. On the other hand, consuming less than 5 portions of fruit and vegetables per day increases the risk of depression, which was shown in the analysis of the diet of people living in Bangladesh, India, and Nepal [260]. Studies in rats have shown that the polyphenol-rich fruits of the *Acacia hydaspica* plant exert antidepressant and anxiolytic effects by improving the antioxidant status in the brain [261]. Banana fruit pulp and peel, which contain polyphenols, have been shown to have both anti-anxiety and anti-depressant effects [262]. *Grewia asiatica* berry juice, rich in phenols, anthocyanins, vitamin C, and flavonoids, improves symptoms of depression by reducing oxidative damage in the brain and increasing levels of SOD and GPx [263]. *Saccharina japonica* ethanol extract reduced depression-like behavior in stressed mice and increased SOD activity [264]. On the other hand, maqui berries improved health in post-stroke depression by increasing the level of GSH expression and increasing the activity of SOD and CAT [265].

A diet rich in flavonoids, consisting of a variety of vegetables, especially yellow, orange, red, and green leafy vegetables, can help reduce symptoms of depression [266]. Numerous studies have also found an increase in antioxidant parameters (SOD, CAT, GPx, Gsr activity, GSH, GSTs, vitamin C, and vitamin E) and a decrease in oxidative stress parameters (TBARS, lipid hydroperoxides, conjugated dienes tissue, and circulatory levels) as a result of the use of flavonoid chrysin [157,159,208,211]. Resveratrol improves antioxidant defense by increasing the activity of CAT, SOD, GPx, and glutathione S-transferase (GST) [145]. Tannic acid increases the level of endogenous antioxidants in the tissues of laboratory animals poisoned with xenobiotics, including the brain [96,202,204,238,267]. Ferulic acid in mice reduced the markers of oxidative stress (MDA, nitrite, and PC) in the brain [268] and increased the activity of SOD, CAT, and GSH-Px in the cerebral cortex and decreased the level of TBARS in mice under stress [269]. In studies on mice, the positive effect of gamma-aminobutyric acid (GABA) from green tea on the course of depression after stroke was shown, and this effect was correlated with the antioxidant activity and phytochemical composition of tea [270,271]. Quercetin was shown to be effective in preventing depressive behaviors, through the reduction of thiols, TBARS, MDA, and NO expression and an increase in the activity of CAT, SOD, and GSH [183].

### 5.4. Neuroprotective Action of Polyphenols

The neuroprotective effect of phenolic compounds is due to their anti-inflammatory and antioxidant properties. Dietary polyphenols may have beneficial effects on health, but their direct effect on neuronal cells is not fully known, as most phenols are metabolized and do not reach the brain in the form in which they are found in food sources [272]. However, a study in mice showed that some amounts of polyphenols are present in the brains of animals that receive polyphenols, suggesting that the polyphenols may act directly in the brain [273]. The results of experimental studies suggest that plant preparations rich in phenolic compounds may be effective in reversing neurodegenerative pathology and age-related declines in neurocognitive performance.

Polyphenols modulate specific cell signaling pathways involved in cognitive processes [136]. Curcumin exerts neuroprotective effects through several signaling pathways, including TLR-4-dependent inflammatory signaling, and may inhibit TLR-4 activation [191]. It also inhibits the production of pro-inflammatory markers in microglial cells, such as cyclooxygenase-2 (COX2), by inhibiting NF-κB, IL-1β, and IL-6 [193]. The general neuroprotective effect of curcumin is a result of the activation of molecular chaperones as well as an increase in the level of neutrophilic factors and the chelation of toxic metals [193]. Terminalia chebula polyphenolic extracts exert a neuroprotective effect in the BV2 microglial cells of the mouse brain and occlusion of the middle artery (in vitro study) by stimulating Nrf2, thus inhibiting apoptosis [274]. Nrf2 translocation from the cytosol to the nucleus promotes the expression of antioxidant genes, including heme oxygenase-1 (HO-1), and increases the activity of antioxidant-related enzymes, including SOD and GSH [275]. Rats with induced neonatal hypoxia and cerebral ischemia whose mothers received polyphenols (resveratrol, pterostilbene, and viniferine) isolated from grapefruit during pregnancy displayed reduced cerebral edema and preservation of motor and cognitive functions, including learning and memory [276]. In vitro studies investigated the effect of polyphenols on H_2_O_2_-induced neuronal apoptosis in human SH-SY5Y neuroblastoma cells [272]. Of the 19 metabolites tested, 3,4-dihydroxyphenylpropionic acid, 3,4-dihydroxyphenylacetic acid, gallic acid, ellagic acid, and urolithin prevented neuronal apoptosis by lowering ROS levels, increasing redox activity, and reducing oxidative stress-induced apoptosis by preventing caspase-3 activation by the mitochondrial apoptotic pathway. Another study demonstrated the efficacy of robinetinidol- (4beta-8) -epigallocatechin 3-O-gallate polyphenols isolated from *Acacia mearnsii* in human SH-SY5Y neuroblastoma cells exposed to acrolein [135]. In this study, in addition to improving the parameters of oxidative stress, inhibition of caspase-3 activation, reduction of NADPH oxidase activity, lipid peroxidation, and reduction of phospho-JNK (c-Jun NH2-terminal kinase), which is known as an apoptotic mediator in induced cell death, were also observed. *Acacia hydaspica* extract reversed cisplatin-induced neurotoxicity in the brain tumor therapy of Sprague Dawley rats by regulating acetylcholinesterase activity, reducing DNA damage, and reducing the level of pro-inflammatory cytokines [277]. Pterostilbene, a phenolic compound found in, e.g., blueberries, showed neuroprotective and anti-inflammatory effects in SH-SY5Y human neuroblastoma cells and RAW 264.7 macrophages exposed to lipopolysaccharide [278]. Phenylpropanoids isolated from raspberry fruit showed neuroprotective activity against the oxidative stress induced by H_2_O_2_ in SH-SY5Y human neuroblastoma cells, which was manifested by selective inhibition of the induction of apoptosis and ROS accumulation and an increase in CAT activity [279]. The polyphenolic extracts from wild blackberries, *Rubus brigantinus* and *Rubus vagabundus*, were shown to have neuroprotective effects in a cellular neurodegenerative model, through the lowering of intracellular ROS levels and modulation of glutathione levels, as well as the activation of caspases [280]. An in vitro study confirmed the neuroprotective effects of β-amyloid polyphenolic extracts from blueberries, black raspberries, cranberries, red raspberries, and strawberries on microglia by scavenging free radicals, capturing reactive carbonyl forms, anti-glycation, and anti-fibrillation effects [246]. The in vitro study also confirmed the protective effect of polyphenol-rich blueberry extract on adult human neuronal progenitor cells [281]. Polyphenol-rich infusions of white, red, black, and green tea increased SOD, CAT, and GPx in the brains of Wistar rats poisoned with cadmium (Cd) and lead (Pb) [282]. The antioxidant properties of various teas are believed to be due to their high content of polyphenols such as catechins, including EGCG and quercetin, theaflavins, thearubigins, and tannins, including tannic acid [91]. EGCG has a multidirectional neuroprotective mode of action, including antioxidant, anti-inflammatory, anti-apoptotic, and anti-amyloidogenic effects [283,284]. Tannins, which are phenolic compounds commonly found in fruits, vegetables, herbs, and tea, have antioxidant properties and neuroprotective effects involving the prevention of the accumulation of nitrites, inhibition of the expression and activity of heme oxygenase 1 (HO-1), and a reduction in the degradation of poly glycohydrolase (ADP-ribose) (PARP) [285]. Tannic acid showed a strong neuroprotective effect in an animal model of stroke (transient occlusion of the middle cerebral artery; tMCAO), as well as Zn^2+^ chelating and antioxidant activity in primary cortical neurons of Sprague-Dawley rats [286]. In addition, tannic acid may soothe brain tissue damage caused by chronic exposure to heavy metals, including Cd, aluminum (Al), and Pb, due to its strong antioxidant and metal chelating properties [71,96,287]. Tannic acid also significantly reduces behavioral disturbances, oxidative damage, and inflammatory responses due to brain injury, possibly related to the activation of PGC-1α and the Nrf2/ARE signaling pathway [288]. Long-term administration of tannic acid has also been shown to improve hypoperfusion-induced motor deficits and memory impairment in a rat model of unilateral carotid occlusion [289]. This mechanism may be related to the inhibition of apoptosis and cell death through the activation of antioxidant pathways, especially the Nrf2 pathway. Administration of gallic acid extract from *Terminalia bellirica* fruit to mice ameliorated chronic mild stress-induced depression-like behavior by reduction of serum corticosterone and acetylcholinesterase (which led to regulation of hyperactivity of the hypothalamic-pituitary-adrenal axis), elevation of neurotransmitters, inhibition of monoamino oxidases (which led to modulation of the monoaminergic system), and mitigation of chronic mild stress-induced oxidative stress and apoptotic cell death [290]. In turn, an alcoholic extract of *Terminalia arjuna* bark showed protective activity against picrotoxin-induced anxiety in mice [291].

## 6. Summary and Perspectives

The results of preclinical studies indicate the potential of phenolic compounds in reducing depressive behaviors by regulating factors related to oxidative stress, neuroinflammation, autophagy, dysregulation of the HPA axis, stimulation of monoaminergic neurotransmission and neurogenesis, and modulation of the intestinal microbiota (Figure 3). Future research should focus on describing the therapeutic and prophylactic mechanisms of consuming phenolic-rich foods, with particular emphasis on their epigenetic mechanisms. This knowledge can contribute to the development of more effective, personalized therapies. In addition, understanding the relationship between the microbiome and the brain is essential to developing microbiota-based therapeutic strategies that can be used in brain disorders. Eating a diet rich in polyphenols could form part of dietary manipulation as a non-invasive, natural, and inexpensive therapeutic agent to support a healthy brain. Evidence suggests that polyphenols may have antidepressant effects.

## Figures and Tables

**Figure 1 ijms-24-02258-f001:**
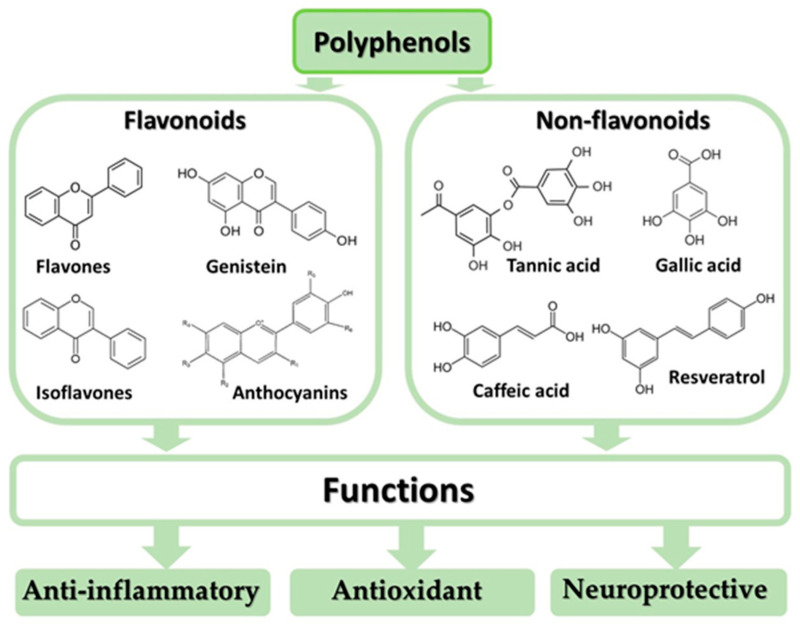
Classification and health benefits of polyphenols.

**Figure 2 ijms-24-02258-f002:**
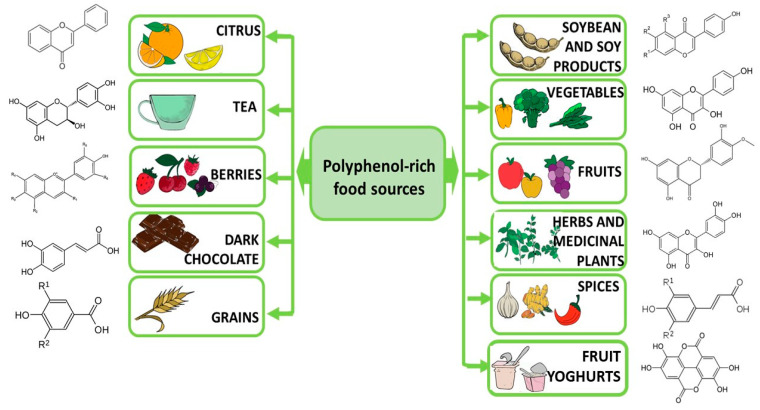
Dietary sources of polyphenols.

**Figure 3 ijms-24-02258-f003:**
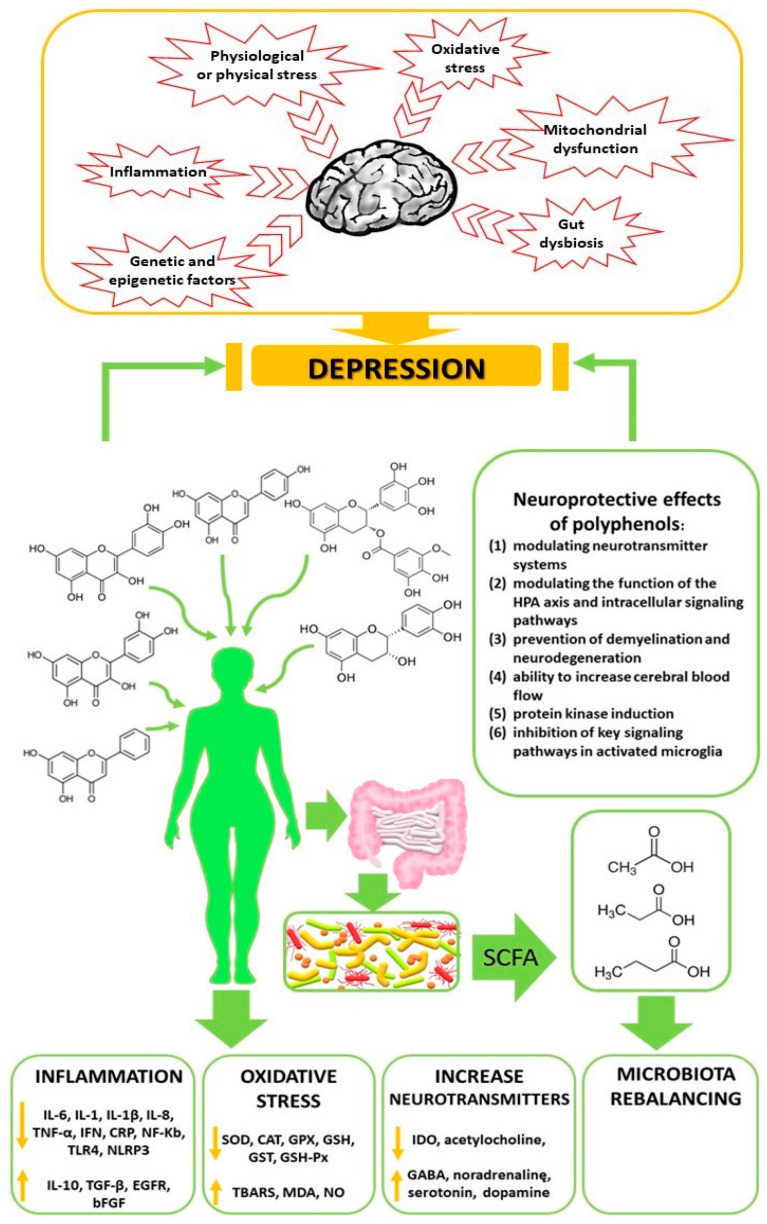
Beneficial effects of dietary polyphenol consumption on the reduction of depression.

**Table 1 ijms-24-02258-t001:** Antioxidant capacity and lipid peroxidation biomarkers in depression—review of studies in humans.

Characteristic	Place of Collection	Antioxidant Capacity	Inflammation Parameters	Peroxidation Biomarkers	References
Control *n* = 20 Depression *n* = 40	Serum	↓ Vitamin A; ↓ Vitamin C; ↓ Vitamin E			[5]
Control *n* = 38 Depression *n* = 42 Generalized anxiety *n* = 37	Plasma	↓ Vitamin E		↑ MDA	[16]
Control *n* = 19 Depression *n* = 15	Serum	↓ GPX; ↓ SOD; ↑ RGSH		↑ MDA	[17]
Control *n* = 12 Depression *n* = 12	Brain	↓ SOD; ↓ CAT; ↓ GPX			[18]
Control *n* = 1484 Depression *n* = 2477	Serum	↑ SOD; ↓ uric acid; ↓ Zn		↑ MDA	[19]
Control *n* = 1788 Depression *n* = 1979	Serum	↑ GST; ↑ CAT; ↓ GSH; ↑ nitrites; ↑ uric acid; ↑ TBARS; ≈SOD; ≈GPX; ≈Zn		↑ MDA	[20]
Control *n* = 36 Depression *n* = 18	Urine	↑ F2 isoprostanes			[21]
Control *n* = 30 Depression *n* = 60	Whole-blood	↓ SOD; ↓ GPX; ↑ adenosine deaminase			[22]
Control *n* = 40 Depression *n* =30	Plasma	↓ GSH; ↓ RGSH; ↓ CAT; ↓ SOD; ≈GPX			[23]
Control *n* = 20 Depression *n* = 58	Serum	≈ SOD; ≈ CAT; ↓ GPX			[24]
Control *n* = 22 Depression *n* =55	Serum	↑ TBARS			[25]
Control *n* = 35 Depression *n* = 73	Serum	↑ TBARS			[26]
Control *n* = 16 Depression *n* = 16	Human dermal fibroblast cultures	↑ PC; ↑ RGSH			[27]
Control *n* = 20 Depression *n* = 20	Plasma		≈8-iso-PGF2a, (+) correlation with IL-6 (–) correlation with IL-10		[28]
Control *n* = 13 Depression *n* = 51 (‘high-risk’, *n* = 15, ‘ultra-high-risk’, *n* = 20, mixed or manic symptoms that received, *n* = 16)	Serum	↓ LPH			[29]
Control *n* = 612 Depression *n* = 2833 (depressive = 1619, remitted = 610, anxiety = 604)	Plasma		↓ 8-OHdG; ≈F2-isoprostanes		[30]
Control *n* = 27 Depression *n* = 22	Serum	↑ PC; ↑ NADPH oxidase; ≈CAT; ≈SOD; ≈GPX			[31]
Control *n* = 10 Depression *n* = 14	RBC	≈SOD1; ≈SOD2; ↑ NADPH oxidase; ↑ ROS + RNS		≈ oxidized LDL	[32]
Depression = 39 Control = 31	Brain	↓ GSH			[33]
Depression = 19 Control = 8	Brain	↓ OCC; ↓ GSH,			[34]
Control *n* = 94 Depression *n* = 55	Serum	↑ PC; ≈TBARS			[35]

↑—Increased concentration or activity in comparison to the control (healthy) group; ↓—decreased or inhibited concentration or activity in comparison to the control (healthy) group; ≈ no differences in comparison to the control (healthy) group; 8-iso-PGF2a—F2alphaisoprostanes; IL-6—intreleukin 6; IL-10—interleukin 10; 8-OHdG—8-hydroxy-2-deoxyguanosine; LDL-chol-low-density lipoprotein cholesterol; F2 isoprostanes—2,3-dinor-5,6-dihydro-15; GSH—glutathione; SOD -superoxide dismutase; CAT—catalase; LPH—lipid hydroperoxides; MDA—malondialdehyde; GST—GSH-transferase; GPX—glutathione peroxidase; PC—protein carbonylation; TBARS—thiobarbituric acid reactive substances; RGSH—reduced glutathione; ROS + RNS—reactive oxygen species + reactive nitrogen species; OCC—occipital cingulate cortex; NADPH—nicotinamide adenine dinucleotide phosphate oxidase; LPH—lipid hydroperoxides.

**Table 2 ijms-24-02258-t002:** The effect of polyphenols on oxidative stress in depression—review of studies in human.

Polyphenols	Oxidative Stress Parameters	Inflammation Parameters	Place of Collection	References
Curcumin	No significant effects on blood chemistry		Blood	[147]
Poliphenols	↓ DTAC; ↓ Vitamin C; ↓ Vitamin A; ↓ Vitamin E		DTAC was calculated	[130]
Curcumin		↓IL-1β, ↓TNF-α; ↓BDNF	Serum	[148]
Curcumin	↑ MDA; ↑ TAC		Serum	[149]

↑—Increased concentration or activity in comparison to the control (healthy) group; ↓—decreased or inhibited concentration or activity in comparison to the control (healthy) group; BDNF—Brain-derived neurotrophic factor; DTAC—dietary total antioxidant capacity; IL-1β—interleukin 1β; MDA—malondialdehyde; TAC—total antioxidant capacity; TNF-α—tumor necrosis factor.

**Table 3 ijms-24-02258-t003:** The effect of polyphenols on oxidative stress in depression—review of studies on laboratory animals.

Polyphenols	Animal Species	Target Sites	Antioxidant Factor	Oxidative Stress Parameters	Inflammation Parameters	Ref.
Curcumin	Sprague- Dawley rats	Hippocampus Serum	CUMS	↓ NOx2; ↓ 4-HNE; ↓ MDA; ↑ CAT		[150]
Curcumin	Rats	Brain	Intracerebroventricular injection of propanoic acid to induce autistic behavior	↓ TBARS; ↑ CAT; ↑ SOD; ↑ GSH		[151]
Curcumin	Male Wistar rats	The cortex and hippocampus	Reserpine treated	↓ MAO		[152]
Curcumin	Male mice	Hippocampus, frontal cortex, amygdala	Regular ICR	↓ MAO-A		[153]
Curcumin	Male rats	Brain	LPS administration		↓COX-2	[154]
Curcumin	Female albino rats	Serum	OVX	↓ MAO-B; ↑ tyrosine Hydroxylase; ↓ NO; ↑ TAC; ↓ MDA	↓IL-1β; ↓IL-6	[155]
Curcumin	Male Swiss mice	Brain	CUMS	↑ CAT		[156]
Chrysin	Mice	Brain	Male young and aged Swiss Albino mice were kept in groups during aging	↑ SOD; ↑ CAT; ↑ GPX in PFC and hippocampus		[157]
Chrysin	Male rats	Heart	Induced acute cardiotoxicity triggered by doxorubicin	↑ GHS; ↑ CAT; ↑ SOD		[158]
Chrysin	Male Swiss mice	Brain	The neurotoxicity elicited by aluminium chloride	↓ LPO; ↑ SOD; ↑ CAT in cortex and hippocampus; ↓ ROS in neuronal SH-SY5Y and microglial THP-1 cells in vitro		[159]
Hypericin	Wistar rats	Brain	Chronic psychosocial stress		↓BDNF	[160]
Dimethyl fumarate	Male Wistar rats	Testicular Serum	CUMS	↓ MDA; ↑ GSH; ↑ TAC		[45]
Dimethyl fumarate and resveratrol	Wistar rats	Brain Serum	CUMS	↓ MDA ↑ TAC; ↑ GSH	↑BDNF	[161]
Resveratrol	Male Wistar rats	Plasma	CUMS	↑ TAC; ↑ GSH	↑ TNF-α; ↑ IL-6; ↑ CRP	[162]
Hesperidin	Male Sprague- Dawley rats	Brain	CUMS	↓ MDA; ↓ nitrite;↑ GSH; ↑ CAT, ↓ NOS;	↓ TNF-α; ↑ IL-1β; ↓ COX-2	[163]
Baicalin	Male Sprague- Dawley rats	Brain	CUMS	↑ SOD; ↓ MDA; ↑ Bcl-2 protein; ↓ Bax	↓ IL-1β; ↓ caspase-3 proteins; ↑ BDNF	[164]
Catechins	Sprague- Dawley rats	Brain	CUMS	↑ CAT; ↑ SOD; ↑ GSH		[165]
Hesperidin	Male mice	Serum	LPS		↓ IL-1β; ↓ IL-6; ↓ TNF-α	[166]
Quercetin	Male Swiss albino mice	Brain, plasma	CUMS	↑ GSH; ↑ SOD;↑ CAT	↓ IL-1β; ↓ TNF-α	[167]
Quercetin	Swiss albino mice	Brain, plasma	CUS	↑ total thiol; ↑ CAT; ↓ TBARS; ↓ NOS;	↓ IL-1β; ↓ IL-6; ↓ TNF-α; ↓ COX-2	[168]
Quercetin	Male Sprague-Dawley rats	Serum	CUMS	↑ SOD; ↑ CAT; ↑ GSH; ↑ GPX; ↓ MAO	↓ IL-1β; ↓ TNF-α	[169]
Quercetin	Male Wistar rats	Brain	CUMS	↓ MDA; ↓ nitrite; ↑ GSH; ↑ SOD	↓ TNF-α; ↓ IL-6; ↓ iNOS	[170]
Ferulic acid	Male mice	Brain	Reserpine	↑ SOD; ↑ GSH; ↓ nitrite; ↓ LPO	↓ IL-1β; ↓ TNF-α	[171]
Salvianolic acid B	Male albino Wistar rats	Brain	CMS	↑ CAT; ↑ SOD; ↑ GPX, ↓ MDA	↓ IL-6; ↓ IL-1β; ↓ TNF-α; ↓ NLRP3	[172]
Resveratrol	Male Wistar rats	Plasma	CUMS	↑ TAC; ↑ GSH	↑ TNF-α; ↑ IL-Iβ; ↑ CRP	[162]
Ginsenoside	Mice	Brain	Behavioral tests (forced swimming, tail suspension)	↑ SOD		[173]
*Apocynum venetum* leaf extract	Wistar rats	Hippocampus, serum	CUMS	↓ ROS; ↓ MDA; ↑ SOD; ↑ CAT; ↑GPX;	↓ Bcl-2/Bax; ↑ BDNF	[174]
*Hemerocallis citrina Baroni*	Sprague- Dawley rats	Brain	CUMS	↓ MDA	↑ BDNF	[175]
*Silybum marianum* (silymarin)	Swiss albino mice	Brain	Acute restraint stress induced by immobilizing	↓ MDA; ↑ SOD; ↑ CAT; ↑ GSH		[176]
*Silybum marianum* (silymarin)	Wistar rats	Brain	Olfactory bulbectomized technique	↓ MDA	↑ BDNF	[177]
Resveratrol	Sprague- Dawley rats	Brain	CUMS	↓ MDA; ↑ SOD; ↑ CAT; ↑ GSH		[178]
*Jasminum sambac* (Jasmine tea)	Sprague- Dawley rats	Brain	CUMS	↑ GLP-1	↑ BDNF	[179]
Tea polyphenols	Mice	Brain	CUMS	↓ MDA; ↑ SOD; ↑ CAT; ↑ GSH		[180]
Okra seeds (catechin and quercetin derivatives)	Mice	Brain	Behavioral tests (open field, tail suspension, forced swimming, novelty suppressed feeding	↓ MDA; ↑ TAC; ↑ SOD		[181]

↑—increased concentration or activity compared to the treated group; ↓—decreased or inhibited concentration or activity compared to the treated group; 4-HNE—4-hydroxynonenal; Bcl-2—B-cell lymphoma 2; Bax—Bcl-2-associated X protein; BDNF—Brain-derived neurotrophic factor; CAT—catalase; COX-2—cyclooxygenase; CRP—C-reactive protein; CUMS—chronic unpredictable mild stress; CUS—chronic unpredictable stress; CMS—chronic mild stress; GLP-1—glucacon-like peptide 1; GSH—glutathione; GPX—glutathione peroxidase; IL-1β—interleukin 1β; IL-6—interleukin 6; iNOS—inducible nitric oxide synthase; LPS—lipopolysaccharide; MAO—monoamine oxidase; MAO–A—monoamine oxidase A; MAO-B—monoamine oxidase B; NOx—index of nitrite/nitrate; LPO—lipid peroxidation; MDA—malondialdehyde; NO—nitric oxide; PFC—prefrontal cortex; ROS—reactive oxygen species; SOD—superoxide dismutase; TBARS—thiobarbituric acid reactive substances; TAC—total antioxidant capacity; TNF-α—tumor necrosis factor. THP-1 monocytic cells; SH-SY5Y—the human neuroblastoma cell line; NLRP3—NOD-like receptor family, pyrin domain containing 3.

## Data Availability

Not applicable.

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
