# Peer review of "Anti-Inflammatory, Antioxidant, and Neuroprotective Effects of Polyphenols—Polyphenols as an Element of Diet Therapy in Depressive Disorders"

_ijms, 2023, doi:10.3390/ijms24032258_

Round 1
Reviewer 1 Report
Dear author, I understand that points 2 and 3 try to explain the pathogenesis of depressive disorders, however I recommend that both points also include polyphenols, so that it is consistent with the title of the article.
Author Response
Reviewer 1
We thank very much Reviewer for the time offered us, the support and comments which were very helpful. We have read comments very carefully and have made correction which we hope meet with Reviewer`s approval. We indexed revisions in yellow color in the manuscript. We respond as follow:
Reviewer`s comment:
Dear author, I understand that points 2 and 3 try to explain the pathogenesis of depressive disorders, however I recommend that both points also include polyphenols, so that it is consistent with the title of the article.
AU: Section 2 considers polyphenols, Section 3 describes only the pathophysiology of depression, while Section 4 describes in detail the antidepressant effects of polyphenols
Thank you for the positive feedback. Thank you very much for such a positive opinion which is very important to us.
Anna Winiarska-Mieczan
Reviewer 2 Report
Winiarska-Mieczan and colleagues, wrote an interesting manuscript reviewing the beneficial effects of polyphenols as an element of diet therapy in depression. The work makes a major emphasis is inflammation, antioxidant capacity and modulation of the gut microbiota as the core of the pathophysiology of depression as well as the therapeutic potential of polyphenols. The manuscript is well written, particularly in terms of the nature of polyphenols, inflammation in general, and molecular pathways associated with antioxidant capacity. Curiously, although this review also focusses on the gut microbiota as an important component of depression and antidepressant action, there are missing some important points related to this work (see my comments below) that I would really encourage to develop in the revision, or at least, a good explanation why the authors think is not relevant for the field. Nevertheless, the manuscript represents a good summary of what is currently known overall. Therefore, I recommend that a major revision is warranted. I explain my concerns in more detail below. I ask that the authors specifically address each of my comments in their response.
- Table 1, title says plasma, but samples vary from plasma, serum, cells etc
- 174-187 when mentioning kyn pathway, a sentence including the reported values in plasma or serum in patients with depression would help to convey the role of this pathway in depression.
- Line 235, “gut barrier dysfunction causes…” better say “gut barrier dysfunction increases the potential influx of…”.
- 229-250, it mentions gut barrier function and defines the consequences of dysfunction, what about BBB function? Is there any correlation between the gut microbiota and BBB?
- 246, please provide a definition for psychobiotic
- Line 249, what is the microbiota-gut-brain axis? Why is relevant for depression and other psychiatric disorders?
- 250, the acronym for HPA is already used in the introduction. Should be abbreviated.
- 253-254, “In inflammatory place as a result…” this sentence seems weird when reading, I suggest revising this part and rewording if necessary.
- 296, model of depression?
- 307, HPA already defined in line 47
- Typo in line 364 “AThe”
- There is not mention of gut microbial diversity whatsoever. What is the current knowledge about alpha or beta diversity when comparing a depressive phenotype vs a healthy condition?
- Maybe it is worth to mention briefly that polyphenols, as vitamins, are obtained mainly through diet, and list typical foods rich in polyphenols as example.
- Line 388, please define a Western diet
- Table 3 is not intuitive to read, the content is ok but I would suggest changing the column order by using the following; Polyphenol, study design (here add species/treatment), target site, oxidative stress parameters, inflammatory parameter, and refs.
- Line 590, definition of probiotic may be required. Same if mentioning prebiotic, psychobiotic, symbiotic, in case they mentioned. The journal is wide scope, and some readers may become confused when reading these concepts.
- Line 636-653, What about fibre in fruits and vegetables? Is not only polyphenolic content. Potential prebiotic effect. Indeed, prebiotics are not mentioned in this review and its potential in modulating the gut microbiota or gut-brain axis. Can polyphenols be considered as prebiotic-like compounds?
- 691 – 694, definition of Nrf2 should be a bit earlier when first mentioned in line 517
- Line 754, better use ‘dysregulation’ instead of ‘deregulation’
Figure 1, is ok, but I would make the box of neuroprotective effects even more brief, particularly in point 2) and 6). In this way it would represent a good summary of the work presented and not falling in too many details
Author Response
Reviewer 2
We thank very much Reviewer for the time offered us, the support and comments which were very helpful. We have read comments very carefully and have made correction which we hope meet with Reviewer`s approval. We indexed revisions in yellow color in the manuscript. We respond as follow:
Reviewer`s comment:
Winiarska-Mieczan and colleagues, wrote an interesting manuscript reviewing the beneficial effects of polyphenols as an element of diet therapy in depression. The work makes a major emphasis is inflammation, antioxidant capacity and modulation of the gut microbiota as the core of the pathophysiology of depression as well as the therapeutic potential of polyphenols. The manuscript is well written, particularly in terms of the nature of polyphenols, inflammation in general, and molecular pathways associated with antioxidant capacity. Curiously, although this review also focusses on the gut microbiota as an important component of depression and antidepressant action, there are missing some important points related to this work (see my comments below) that I would really encourage to develop in the revision, or at least, a good explanation why the authors think is not relevant for the field. Nevertheless, the manuscript represents a good summary of what is currently known overall. Therefore, I recommend that a major revision is warranted. I explain my concerns in more detail below. I ask that the authors specifically address each of my comments in their response.
- Table 1, title says plasma, but samples vary from plasma, serum, cells etc
AU: Thank Reviewer for this highly relevant suggestion. The title of the Table 1 has been changed to „Antioxidant capacity and lipid peroxidation biomarkers in depression - review of studies in human”
- 174-187 when mentioning kyn pathway, a sentence including the reported values in plasma or serum in patients with depression would help to convey the role of this pathway in depression.
AU: Thank Reviewer for this highly relevant suggestion. I wrote: „In plasma, lower levels of picolinic acid, higher levels of quinolinic acid, and reduced levels of neuroprotective to neurotoxic metabolite ratios were found in depressive patients compared to the healthy controls. In the cerebrospinal fluid, a significantly lower level of picolinic acid was found in depressive compared to the healthy subjects [64].” (line 168-172)
- Line 235, “gut barrier dysfunction causes…” better say “gut barrier dysfunction increases the potential influx of…”.
AU: Modified as suggested
- 229-250, it mentions gut barrier function and defines the consequences of dysfunction, what about BBB function? Is there any correlation between the gut microbiota and BBB?
AU: Thank Reviewer for this highly relevant suggestion. I wrote: „SCFAs, i.e. the main metabolites produced in the colon by gut microbiota, are transported via blood vessels to the brain, where they modulate functions of neurons, microglia, and astrocytes and affect the blood-brain barrier (BBB) [80,86]. Elevated levels of toxins and/or microbes may alter the functioning of the BBB, which that may lead to neurodegeneration [80].” (line 247-251)
- 246, please provide a definition for psychobiotic
AU: Thank Reviewer for this highly relevant suggestion. I wrote: „On the other hand, microorganisms belonging to the group of psychobiotics (probiotics and prebiotics which confer mental health benefits through interactions with commensal gut bacteria) produce neurotransmitters, including gamma-aminobutyric acid, serotonin, dopamine, and short-chain fatty acids SCFA (acetic, propionic, and butyric), which directly affect the nervous system [70,88,89].” (line 242-246)
- Line 249, what is the microbiota-gut-brain axis? Why is relevant for depression and other psychiatric disorders?
AU: I wrote: „In this context, the influence of the microbiome-gut-brain axis on the stress response of the hypothalamic-pituitary-adrenal axis in alcohol-induced depression is known [90].” (line 251-252)
- 250, the acronym for HPA is already used in the introduction. Should be abbreviated.
AU: Modified as suggested
- 253-254, “In inflammatory place as a result…” this sentence seems weird when reading, I suggest revising this part and rewording if necessary.
AU: I wrote: „At the site of inflammation, as a result of incomplete reduction of oxygen, among others, superoxide anion is formed which influences the development of oxidative stress.” (line 255-257)
- 296, model of depression?
AU: Modified as suggested
- 307, HPA already defined in line 47
AU: Modified as suggested
- Typo in line 364 “AThe”
AU: Modified as suggested
- There is not mention of gut microbial diversity whatsoever. What is the current knowledge about alpha or beta diversity when comparing a depressive phenotype vs a healthy condition?
AU: I wrote: „Moreover, alpha diversity was found to be negatively associated with depressive symptoms, while beta diversity showed a significant association with major depressive disorder, psychosis, and schizophrenia [73,74].” (line 207-209)
- Maybe it is worth to mention briefly that polyphenols, as vitamins, are obtained mainly through diet, and list typical foods rich in polyphenols as example.
AU: I wrote: „Polyphenols are naturally occurring compounds; they are secondary metabolites of plants. Fruits, vegetables, cereals, and such beverages as tea represent the main sources of polyphenols (Figure 1).”
- Line 388, please define a Western diet
AU: I wrote: „It has also been confirmed that Western eating styles (low in fruits and vegetables, high in fat, saturated fatty acids, sugar, sodium, and processed food) can increase the risk and severity of depression in adolescents [145].” (line 291-293)
- Table 3 is not intuitive to read, the content is ok but I would suggest changing the column order by using the following; Polyphenol, study design (here add species/treatment), target site, oxidative stress parameters, inflammatory parameter, and refs.
AU: Modified as suggested
- Line 590, definition of probiotic may be required. Same if mentioning prebiotic, psychobiotic, symbiotic, in case they mentioned. The journal is wide scope, and some readers may become confused when reading these concepts.
AU: Modified as suggested
- Line 636-653, What about fibre in fruits and vegetables? Is not only polyphenolic content. Potential prebiotic effect. Indeed, prebiotics are not mentioned in this review and its potential in modulating the gut microbiota or gut-brain axis. Can polyphenols be considered as prebiotic-like compounds?
AU: Thank Reviewer for this highly relevant suggestion. I wrote: „In this regard, research suggests that polyphenols are candidates for prebiotic-like compounds. Prebiotics are non-living gut microbial stimulants.” (line 279-281)
- 691 – 694, definition of Nrf2 should be a bit earlier when first mentioned in line 517
AU: Modified as suggested
- Line 754, better use ‘dysregulation’ instead of ‘deregulation’
AU: Modified as suggested
Figure 1, is ok, but I would make the box of neuroprotective effects even more brief, particularly in point 2) and 6). In this way it would represent a good summary of the work presented and not falling in too many details
AU: Modified as suggested
Thank you for the positive feedback. Thank you very much for such a positive opinion which is very important to us.
Anna Winiarska-Mieczan
Reviewer 3 Report
Firstly,
I congratulates to all the authors for the keen interest on a predominant topic in our day-to-day life.
I am happy to accept the review paper in the present form.
The introduction part is quite interesting, which showed a detailing of the anti-oxidants, plasma biomarkers in human.
More details of the phytocompounds was missing.
Please include few more references with respect to plant compound role in neuroprotection and neurodegenerative diseases. Some reference such as .... 1. Yadavalli, C., Garlapati, P.K. & Raghavan, A.K. Gallic Acid from Terminalia Bellirica Fruit Exerts Antidepressant-like Activity. Rev. Bras. Farmacogn. 30, 357–366 (2020)
2. Sekhar, Y. C., Kumar, G. P., & Anilakumar, K. R. (2017). Terminalia arjuna bark extract attenuates picrotoxin-induced behavioral changes by activation of serotonergic, dopaminergic, GABAergic and antioxidant systems. Chinese journal of natural medicines, 15(8), 584-596.
Author Response
Reviewer 3
We thank very much Reviewer for the time offered us, the support and comments which were very helpful. We have read comments very carefully and have made correction which we hope meet with Reviewer`s approval. We indexed revisions in yellow color in the manuscript. We respond as follow:
Reviewer`s comment:
Firstly,
I congratulates to all the authors for the keen interest on a predominant topic in our day-to-day life.
I am happy to accept the review paper in the present form.
The introduction part is quite interesting, which showed a detailing of the anti-oxidants, plasma biomarkers in human.
More details of the phytocompounds was missing.
Please include few more references with respect to plant compound role in neuroprotection and neurodegenerative diseases. Some reference such as .... 1. Yadavalli, C., Garlapati, P.K. & Raghavan, A.K. Gallic Acid from Terminalia Bellirica Fruit Exerts Antidepressant-like Activity. Rev. Bras. Farmacogn. 30, 357–366 (2020)
- Sekhar, Y. C., Kumar, G. P., & Anilakumar, K. R. (2017). Terminalia arjuna bark extract attenuates picrotoxin-induced behavioral changes by activation of serotonergic, dopaminergic, GABAergic and antioxidant systems. Chinese journal of natural medicines, 15(8), 584-596.
AU: As suggested by the Reviewer, literature has been cited . I wrote: „Administration of gallic acid extract from Terminalia bellirica fruit to mice ameliorated chronic mild stress-induced depression-like behavior by reduction of serum corticosterone and acetylcholinesterase (which led to regulation of hyperactivity of the hypothalamic-pituitary-adrenal axis), elevation of neurotransmitters, inhibition of monoamino oxidases (which led to modulation of the monoaminergic system), and mitigation of chronic mild stress-induced oxidative stress and apoptotic cell death [290]. In turn, alcoholic extract of Terminalia arjuna bark showed protective activity against picrotoxin-induced anxiety in mice [291].”
Thank you for the positive feedback. Thank you very much for such a positive opinion which is very important to us.
Anna Winiarska-Mieczan
Reviewer 4 Report
1: Minor grammatical corrections throughout the text required.
2: Botanical/microbial nomenclature and In-vitro and In-vivo must be italic; correction required throughout the text.
3: Tables and Figure need revision to make more informative, presentable and understandable.
4: A Table may be included containing dietary/natural sources of polyphenols, particularly those discussed in the article. It will enhance the understanding level of readers and justify the title.
5: "Yogurt is a rich source of antioxidants also containing polyphenols in addition to probiotic effect. Presence of dairy products enhance the stability and efficacy of polyphenols" Special attention required.
6: Majority headings comprise a group discussion on multiple molecules which often lead to confuse the reader, splitting the discussion will make it more clear.
Author Response
Reviewer 4
We thank very much Reviewer for the time offered us, the support and comments which were very helpful. We have read comments very carefully and have made correction which we hope meet with Reviewer`s approval. We indexed revisions in yellow color in the manuscript. We respond as follow:
Reviewer`s comment:
1: Minor grammatical corrections throughout the text required.
AU: Modified as suggested
2: Botanical/microbial nomenclature and In-vitro and In-vivo must be italic; correction required throughout the text.
AU: Modified as suggested
3: Tables and Figure need revision to make more informative, presentable and understandable.
AU: Modified as suggested. Table 3 was rebuilt, the order of columns in it was changed - now it is more readable, in fig. 1 the description of Neuroprotective effects of polyphenols was shortened
4: A Table may be included containing dietary/natural sources of polyphenols, particularly those discussed in the article. It will enhance the understanding level of readers and justify the title.
AU: Thank Reviewer for this highly relevant suggestion. The information is shown in Fig. 1
5: "Yogurt is a rich source of antioxidants also containing polyphenols in addition to probiotic effect. Presence of dairy products enhance the stability and efficacy of polyphenols" Special attention required.
AU: I wrote: „It is also recommended that products containing polyphenols be included in the diet under certain conditions. A good example is yoghurt. Enriched with plant extracts, it makes a very good source of polyphenols and other antioxidants, and can additionally contain probiotic bacteria having a positive effect on the intestinal microbiome [256].” (line 627-630)
6: Majority headings comprise a group discussion on multiple molecules which often lead to confuse the reader, splitting the discussion will make it more clear.
AU: Modified as suggested. Chapter 5 is divided into sub-chapters.
Thank you for the positive feedback. Thank you very much for such a positive opinion which is very important to us.
Anna Winiarska-Mieczan